# Percutaneous Endoscopic Necrosectomy—A Review of the Literature

**DOI:** 10.3390/jcm11143932

**Published:** 2022-07-06

**Authors:** Mateusz Jagielski, Agata Chwarścianek, Jacek Piątkowski, Marek Jackowski

**Affiliations:** Department of General, Gastroenterological and Oncological Surgery Collegium Medicum, Nicolaus Copernicus University, 87-100 Torun, Poland; agata.chwarscianek@icloud.com (A.C.); jpiatkowski@wp.pl (J.P.); jackowscy@hotmail.com (M.J.)

**Keywords:** necrosis, retroperitoneal access, transmural access, preferred

## Abstract

In this article, an attempt was made to clarify the role of percutaneous endoscopic necrosectomy (PEN) in the interventional treatment of pancreatic necrosis. A comprehensive review of the current literature was performed to identify publications on the role of PEN in patients with consequences of acute necrotizng pancreatitis. The aim of the study was to review the literature on minimal invasive necrosectomy, with emphasis on PEN using esophageal self-expanding metal stents (SEMS). The described results come from 15 studies after a review of the current literature. The study group comprised 52 patients (36 men and 16 women; mean age, 50.87 (13–75) years) with walled-off pancreatic necrosis, in whom PEN using a self-expandable esophageal stent had been performed. PEN was successfully completed in all 52 patients (100%). PEN complications were observed in 18/52 (34.62%) patients. Clinical success was achieved in 42/52 (80.77%) patients, with follow-up continuing for an average of 136 (14–557) days. In conclusion, the PEN technique is potentially effective, with an acceptable rate of complications and may be implemented with good clinical results in patients with pancreatic necrosis.

## 1. Introduction

A step-up approach consisting of gradual broadening of access to necrotized areas using endoscopic techniques involving a transmural approach or surgical techniques involving a percutaneous approach is recommended in the treatment of the consequences of acute necrotizing pancreatitis [1,2]. The step-up approach using minimally invasive techniques has been shown to improve the results of treatment and reduce complications compared to the traditional, open necrosectomy-based management in patients with pancreatic necrosis [3]. Recent randomized clinical trials comparing both minimally invasive approaches (the endoscopic step-up approach and the surgical step-up approach) have demonstrated that the endoscopic approach is associated with a lower incidence of pancreatic fistulas and a shorter duration of hospital stay, making endoscopic drainage the procedure of choice in the minimally invasive management of pancreatic necrosis [4,5].

However, just like any other technique, endotherapy for the consequences of acute necrotizing pancreatitis has limitations. The main limitation of endoscopic techniques that utilize transmural access in the treatment of pancreatic necrosis is the distance between the gastrointestinal lumen and the necrotic collection, which should not exceed 40 mm, as visualized on endosonographic scans [6,7,8]. If the distance is longer than 40 mm, it is technically impossible to access the necrotic collection through the transmural route [6,7,8,9,10]. In these cases, the surgical step-up approach involving percutaneous drainage from the retroperitoneal access remains the treatment of choice [3,4,5,11].

## 2. The Strategy of Minimally Invasive Treatment for Pancreatic Necrosis

### 2.1. Surgical Step-Up Approach

#### 2.1.1. Percutaneous Drainage

Percutaneous drainage consists of the insertion of a drainage tube (10–28 Fr) into the necrotic collection from a transperitoneal or retroperitoneal access, under ultrasound or computed tomography guidance, with the flushing of the necrotic cavity using physiological saline at intervals of approximately 8 h [12,13,14]. Retroperitoneal access to the necrotic collection is preferred [14]. The transperitoneal approach was used in selected cases when the retroperitoneal approach was technically infeasible [14].

Percutaneous drainage of infected pancreatic necrosis was first reported in 1998 by Freeny et al., who had used this approach on 34 patients [11,13]. Favorable outcomes were achieved in 47% of patients, with a complication rate of 71% and a mortality rate of 12% [13]. In a meta-analysis of eight studies involving 286 patients with infected pancreatic necrosis, percutaneous drainage, which was considered the only means of accessing the necrotic collection, was effective in 44% of patients [15]. However, morbidity and mortality rates were as high as 28% and 20%, respectively. In a meta-analysis of 11 studies involving 384 patients, percutaneous drainage was an effective method of treatment in more than half (55.7%) of the patients, with the requirement of no other interventional treatment [16,17].

When ongoing percutaneous drainage proves ineffective and interventional treatment is still required, continuing the drainage facilitates the postponement of the intervention until the patient’s clinical condition improves and better containment of necrotic collection is achieved [12,18]. As shown in numerous studies, the curative effects of percutaneous drainage, as the only means of accessing the necrotic tissue with no need to resort to other interventional treatments, were achieved in 23–47% of patients [3,19,20]. However, if interventional treatment is still required despite percutaneous drainage of pancreatic necrosis, necrosectomy or mechanical removal of necrotic tissue from the necrotic collection using various surgical tools can be performed using percutaneous access [19,20,21,22,23,24,25].

#### 2.1.2. Retroperitoneal Necrosectomy

Retroperitoneal necrosectomy was first described in 1998 by Gambiez et al. [20]. In 20 patients with pancreatic necrosis, a 6 cm incision was made in the lumbar region and retroperitoneal necrosectomy was performed under mediastinoscopic guidance [20]. The treatment was successful in 85% of patients [20].

Minimally invasive techniques utilizing peritoneal access that have gained widespread recognition include sinus tract endoscopy [21] (also referred to as minimally invasive retroperitoneal pancreatic necrosectomy (MIRPN) [22] or minimal access retroperitoneal pancreatic necrosectomy (MARPN) [23]) and video-assisted retroperitoneal debridement (VARD) [24]. In the aforementioned techniques, retroperitoneal access to the necrotic collection was achieved by means of a percutaneous drainage tube, which was placed under radiological imaging guidance [25]. Thus, the first step in all the aforementioned techniques is percutaneous drainage, with subsequent retroperitoneal necrosectomy only in cases where the drainage is ineffective [3,4,5,26].

The sinus tract endoscopy technique was first described in 2000 by Carter et al. [21]. The same technique was presented in 2003 by Connor et al. under the name “minimally invasive retroperitoneal pancreatic necrosectomy” [22]. In 2010, the term “minimal access retroperitoneal pancreatic necrosectomy” was coined by Raraty et al. to refer to the same technique [23]. Sinus tract endoscopy consists of progressive broadening of the channel formed upon the insertion of a percutaneous drainage tube until a diameter of 30 Fr is reached. A rigid nephroscope [22,23] or a flexible endoscope [21] is then inserted into the necrotic cavity to enable flushing or aspiration of the necrotic contents [21,22,23]. Necrotic tissues are removed using various endoscopic instruments [21,22,23]. Normally, three to five necrosectomy procedures, as described above, are sufficient for the complete evacuation of the necrotic collection [21,22,23]. Clinical success was achieved in 75–86% of patients who were treated using this method [21,22,23]. Complications were observed in 25–88% of patients, and the mortality rates ranged from 0 to 25% [21,22,23,27].

Sinus tract endoscopy may be used as a supportive treatment following open necrosectomy [28]. Castellanos et al. presented the treatment results of 11 patients in whom open necrosectomy was performed via retroperitoneal access [28]. Drainage tubes (18 Fr and 32 Fr) were retained after surgery for postoperative flushing [28]. After the surgery, a fiberoscope was introduced into the necrotic collection to remove the necrotic debris [28]. Necrosectomy procedures were repeated until the necrotic tissue was completely removed, and the procedures could be performed at the bedside without the need for repeated surgery [28].

Another widely recognized technique for the minimally invasive treatment of pancreatic necrosis using retroperitoneal access is VARD, a hybrid between sinus tract endoscopy and open retroperitoneal necrosectomy [18]. The technique was first described by Horvath et al. in 2001 as “laparoscopic assisted percutaneous drainage” [29] and was propagated by van Santvoort et al. in 2007 [24]. In VARD, a 5–6 cm incision is made near the insertion of the percutaneous drainage tube, with the removal of liquid contents using an aspiration system and the removal of solid necrotic tissue using long forceps [19,24,29]. The procedure was performed under the guidance of a camera, which was introduced into the necrotic cavity through a laparoscopic port [19,24,29]. Carbon dioxide was administered for the necrotic collection via a percutaneous drain tube [19,24,29]. After treatment, two large-diameter drainage tubes were retained in the cavity to facilitate postoperative drainage [19,24,29]. This technique facilitates the removal of necrotic tissue in a single procedure [12,13] and was effective in 60–93% of cases [19,24,29]. However, this technique is burdened by a high risk of complications (24–54% of patients), with a mortality rate of 0–8% [19,29,30].

### 2.2. Endoscopic Step-Up Approach

#### 2.2.1. Transmural Drainage

Transmural endoscopic drainage is the main element in the endoscopic management of walled-off pancreatic necrosis as an effect of late-phase acute necrotizing pancreatitis [12,31,32]. Endoscopic transmural drainage consists of the formation of a fistula between the lumen of the pancreatic necrosis collection and the lumen of the gastrointestinal tract to facilitate the free flow of necrotic contents into the gastrointestinal tract [6,7,8,12,31,32]. During endoscopic transmural drainage, transmural puncture of the pancreatic necrotic collection is performed under endoscopic ultrasound guidance, provided that the distance between the wall of the gastrointestinal tract (stomach or duodenum), as visualized on the endosonographic scan, does not exceed 40 mm [6,7,8]. The puncture is then broadened using a cystostome, with the formation of a transmural fistula; that is, a connection is established between the upper gastrointestinal tract (stomach or duodenum) and the necrotic collection [6,7,8]. The next step in the endoscopic procedure is the enlargement of the pancreaticogastric or pancreaticoduodenal fistula [6,7,8]. Following the enlargement, a self-expanding transmural metal stent or plastic endoprosthesis(-es) is placed to facilitate free passive drainage of the collection contents into the gastrointestinal tract [6,7,8]. In cases of post-inflammatory necrotic collections containing necrotic tissue in addition to liquefied necrotic contents, active transmural drainage is required, and a nasal drain is additionally inserted across the fistula to flush the collection during the post-procedural period [6,7,8].

#### 2.2.2. Direct Endoscopic Necrosectomy from Transmural Access

Despite its high efficacy, endoscopic drainage is burdened by limitations, one of which is its limited ability to evacuate poorly liquefied necrotic matter. The next stage of the endoscopic step-up approach, which uses transmural access, consists of direct endoscopic necrosectomy, which is performed during the course of transmural drainage. The procedure involves the insertion of the endoscope along the fistula into the collection lumen and the mechanical removal of necrotic debris using various endoscopic instruments [1,2,4,5,6,7,8,12,31,32]. Indications for endoscopic necrosectomy during the course of transmural drainage of pancreatic necrosis include infection of necrotic tissues or inefficacy of transmural drainage.

The first report of endoscopic necrosectomy was published in 2000 [33]. Seifert et al. presented the treatment outcomes of three patients with pancreatic necrosis in whom necrotic debris was removed using a Dormia basket attached to a gastroscope, which was inserted into the collection lumen [33]. Since then, there has been increased interest and significant development of this technique for the management of pancreatic necrosis [1,2,4,5,6,7,8,12,31,32]. In 2005, Seewald et al. presented the treatment results of 13 patients with pancreatic necrosis who underwent endoscopic necrosectomy, with complete regression of the necrotic collection achieved in 11/13 (85%) patients and the average numbers of necrosectomy and endoscopic lavage procedures amounting to 7 (2–23) and 12 (2–41), respectively [34]. Treatment complications were observed in 4/13 (30.77%) patients [34]. In a subsequent study by Seifert et al., favorable treatment outcomes were achieved in 75/93 (81%) patients, with the mean number of necrosectomies amounting to 6.2 (1–35) [35]. Complications were reported in 24/93 (26%) patients [35]. In a multicenter study by Gardner et al., treatment was successful in 91% of patients [36]. Comparable outcomes (19/22, 86%) were achieved by Risch et al. [37]. In both studies, the average number of endoscopic procedures was four [36,38]. Gardner et al. reported a complication rate of 14% [36]. Risch et al. observed complications in 13% of patients [38].

In their meta-analysis of endoscopic necrosectomy reports, van Brunschot et al. showed that treatment was successful in 81% of 455 patients, with the average number of procedures amounting to 4 (1–23) [39]. Treatment complications were observed in 36% of patients [39]. In their multicenter, randomized study, Bakker et al. demonstrated that the risk of organ lesions, systemic complications, and deaths in patients with infected pancreatic necrosis who underwent endoscopic transmural necrosectomy was lower than that in patients who underwent surgical necrosectomy [40].

#### 2.2.3. Percutaneous Endoscopic Necrosectomy (PEN)

We have presented two different strategies for a minimally invasive step-up approach in patients experiencing the consequences of acute necrotizing pancreatitis, namely the surgical and endoscopic step-up approaches [4,5]. According to a common belief, the surgical step-up approach is the standard treatment, with the endoscopic step-up approach acting as a promising alternative.

The surgical step-up approach is based on external drainage means, that is, percutaneous drainage of a pancreatic necrosis, which is the first step in the therapeutic ladder, leading to the broadening of the access and retroperitoneal necrosectomy [3,26,41]. In contrast, the endoscopic step-up approach is based on internal drainage, that is, transmural drainage of necrotic contents [1,2,4,5,6,7,8]. The advantage of internal (endoscopic) drainage over external (percutaneous) drainage is mainly associated with a lower risk of infection. Furthermore, endotherapy entails no risk of pancreaticocutaneous fistula formation, as opposed to percutaneous drainage [3,4,5,26,41]. On the other hand, percutaneous access facilitates drainage, regardless of the location of the necrotic collection, and percutaneous drainage is the least invasive of all minimally invasive methods that can be performed at the bedside without the need for general anesthesia, which is very important for patients with severe clinical conditions and a high risk of perioperative death or anesthesia-related complications [3,26,41].

The choice of management method for pancreatic necrosis using minimally invasive techniques should be guided by the experience of the treatment center (Figure 1) [6,7,8]. In recent years, the endoscopic step-up approach has been shown to be advantageous over the surgical step-up approach in terms of the incidence of pancreatic fistulas and the duration of hospital stay [4,5], making endoscopic drainage the preferred treatment technique [1]. However, just like any other technique, endotherapy for the consequences of acute necrotizing pancreatitis has limitations [1,4,5,6,7,8]. The main limitation of endoscopic techniques that utilize transmural access in the treatment of pancreatic necrosis is the distance between the gastrointestinal lumen and the necrotic collection, which should not exceed 40 mm on endosonographic scans [6,7,8]. If the distance is longer than 40 mm, it is technically impossible to access the necrotic collection through the transmural route [6,7,8]. In these cases, percutaneous drainage of the necrotic collection, with potential escalation to surgical necrosectomy, remains the treatment of choice [3,19,20,21,22,23,24,25,26,27,28,29,30]. However, isolated cases of an alternative approach that combines the elements of the endoscopic step-up approach with those of the surgical step-up approach and is referred to as PEN have been published recently [42,43,44,45,46,47,48].

In 2011, Bakken et al. [42,43] presented a new method for the minimally invasive treatment of pancreatic necrosis. This method involves endoscopic techniques and percutaneous retroperitoneal access. PEN [42,43,44,45,46,47,48] consists of the gaining of access to the necrotic collection via percutaneous drainage, followed by broadening of the access and percutaneous placement of a large-diameter self-expandable esophageal stent to facilitate percutaneous insertion of an endoscope into the necrotic collection for endoscopic necrosectomy. Only a few papers [37,44,45,46,47,48,49,50,51,52,53,54,55] have presented the treatment outcomes of patients with pancreatic necrosis who underwent PEN since the first published description [42,43] of the technique. This paper reviews the available literature to evaluate the efficacy and safety of PEN for the treatment of patients with acute necrotizing pancreatitis.

## 3. Materials and Methods

The aim of the study is to review the literature on minimally invasive necrosectomy with emphasis on PEN using esophageal self-expanding metal stents (SEMS).

### 3.1. Literature Search

A systematic review was performed by searching Medline, PubMed and Embase databases for articles published between January 2011 and December 2021 that contain the following keywords: “esophageal stent” or “esophageal endoprosthesis” AND “necrosectomy” AND “acute pancreatitis” or “pancreatic necrosis”.

### 3.2. Eligibility Criteria

Publications where the study group fulfilled the following criteria.

#### 3.2.1. Study Inclusion Criteria

All patients with pancreatic necrosis due to acute or chronic pancreatitis were enrolled. The patients underwent PEN with use of esophageal SEMS. All patients aged over 18 years old were included.

#### 3.2.2. Study Exclusion Criteria

Patients with pancreatic necorisis that were not a consequence of pancreatic inflammatory disease were excluded from the study. The study also excluded patients with post-inflammatory PPFCs without clinical symptoms and those who had undergone surgery in the pancreatic region. Moreover, patients aged under 18 years old and pregnant women were excluded.

### 3.3. Data Extraction and Quality Assessment

Two authors independently selected publications based on the inclusion criteria, documented the methodological quality, assessed risks of bias in the included studies and extracted data. The quality assessment was limited by small amount of data and small number of available articles on this issue (Table 1).

In total, 52 patients from 15 studies were included in this review [37,42,43,44,45,46,47,48,49,50,51,52,53,54,55] (Table 1).

### 3.4. The Detailed Description of PEN Technique

The PEN technique [37,42,43,44,45,46,47,48,49,50,51,52,53,54,55] (Figure 1A–M) consists of percutaneous puncture of the pancreatic necrotic collection from retroperitoneal, or less frequently, transperitoneal access, under ultrasound or CT guidance. Next, a fully-coated, self-expandable esophageal stent is inserted across the puncture, with its distal end reaching the necrotic collection lumen and its proximal end located outside the patient’s body. During PEN, a flexible endoscope (usually a gastroscope) is inserted along the esophageal stent lumen and necrosectomy (mechanical evaluation of necrotic debris from the cavity) is performed using various endoscopic instruments. During the procedure, the cavity is extensively flushed, usually with physiological saline, and the contents are aspirated. If subsequent PEN procedures are required, the esophageal stent is retained in the percutaneous location, with the introduction of plastic endoprostheses or drainage tubes along its lumen for passive or active post-procedural drainage of the necrotic collection, respectively. After the completion of endoscopic treatment using percutaneous access, the esophageal stent is removed and the site is secured with an ostomy pouch to collect the residual contents of the necrotic cavity.

### 3.5. Statistical Analysis

All statistical calculations were conducted using the statistical software TIBCO Software Inc. (Palo Alto, CA, USA) (2017). Statistica software (Data Analysis Software System), version 13, was also used (http://statistica.io, accessed on 1 January 2022). Quantitative variables are characterized using arithmetic means, standard deviation, median, and minimum and maximum values (range). Qualitative variables are presented as numbers and percentages.

## 4. Results

The study group comprised 52 patients (36 men and 16 women; mean age, 50.87 (13–75) years) with walled-off pancreatic necrosis in whom PEN using a self-expandable esophageal stent had been performed.

Choledocholithiasis, as observed in 19/52 (36.54%) patients, was the most common etiology of acute pancreatitis leading to the walled-off pancreatic necrosis. The remaining, less common causes of acute pancreatitis in the study group included hyperlipidemia (12 patients), alcohol abuse (6 patients), iatrogenic causes (6 patients), and autoimmune causes (1 patient); no cause of acute pancreatitis was identified in 8 patients.

The average size of the walled-off pancreatic necrosis collection was 19.98 (11.2–33.44) cm. Infection of necrotic areas was observed in 35/52 (67.31%) patients.

PEN was performed via an esophageal stent, which was inserted percutaneously into the necrotic region in all 52 patients. The stent was usually introduced 13.6 (1–37) days into the interventional treatment of the pancreatic necrosis. The average stent length was 126 (80–180) mm, and the average diameter was 20 (18–22) mm. The Dormia basket and, less frequently, the polypectomy loop, were the endoscopic tools used for the removal of necrotic contents. The average number of PEN procedures in one patient was 3.89 (1–7). In most patients, plastic endoprostheses were left in place between the PEN procedures or drainage tubes were inserted into the necrotic collection along the esophageal stent lumen. PEN was successfully completed in all 52 patients (100%). In 19/52 (36.54%) patients, additional transmural endoscopic access using a lumen-apposing metal stent (LAMS) was required.

PEN complications were observed in 18/52 (34.62%) patients. Eleven patients required treatment due to bleeding into the collection lumen. A pancreaticocutaneous fistula was observed in seven patients. The periprocedural mortality rate was 1.92% (one patient).

Esophageal stents were retained in the body for an average of 19.61 (2–48) days during the course of treatment. The mean time between stent removal and closure of the cutaneous fistula was 32.24 (11–92) days.

Clinical success was achieved in 42/52 (80.77%) patients, with follow-up continuing for an average of 136 (14–557) days.

## 5. Discussion

As demonstrated in this study, PEN is a hybrid technique for the minimally invasive treatment of pancreatic necrosis that combines endotherapy with percutaneous access methods [37,42,43,44,45,46,47,48,49,50,51,52,53,54,55]. PEN is effective and safe in cases where transmural endoscopic drainage is technically infeasible [37,42,43,44,45,46,47,48,49,50,51,52,53,54,55]. Moreover, in selected patients with pancreatic necrosis, PEN can be performed in the course of transmural endoscopic drainage when the necrosis is spreading toward the pelvis and paracolic gutters, as transmural endoscopic drainage was shown to be associated with poorer outcomes in these cases [37,42,43,44,45,46,47,48,49,50,51,52,53,54,55]. Thus, PEN improves the outcomes of endoscopic treatment [37,42,43,44,45,46,47,48,49,50,51,52,53,54,55].

The combination of endoscopic and percutaneous drainage techniques was first described in 2010 by Ross et al., who published the treatment results of 15 patients with symptomatic walled-off pancreatic necrosis who underwent combined endoscopic and percutaneous drainage [56]. The authors demonstrated favorable clinical outcomes with low procedure-related morbidity and mortality rates [52,56]. In subsequent publications, the same authors referred to the combined endoscopic and percutaneous drainage as dual-modality drainage [57,58]. In 2011, Gluck et al. demonstrated that, compared to standard percutaneous drainage, dual-modality drainage reduced the rates of interventional procedures and hospitalization and treatment times in patients with pancreatic necrosis [57]. In 2014, Ross et al. reported the treatment outcomes of 117 patients with symptomatic walled-off pancreatic necrosis in whom dual-modality drainage had been performed [58]. Treatment success was achieved in 88.03% of patients, and complications were observed in 5.13% of patients [58]. A study published in 2018 presented the results of dual-modality drainage management in 20 patients with symptomatic pancreatic necrosis [14]. Therapeutic success was reported in 90% of patients, with complications observed in 20% of patients [14].

The dual-modality drainage technique (i.e., the combination of endoscopic and percutaneous drainage of pancreatic necrosis) involves percutaneous drainage, followed by transmural endoscopic drainage [56,57,58]. Thus, the dual-modality drainage approach to the management of pancreatic necrosis is based on percutaneous drainage, with passive endoscopic transmural drainage being performed at a later stage [56,57,58]. In our institution, we tend to avoid performing percutaneous drainage of pancreatic necrosis prior to endoscopic drainage [7,8,14]. Early percutaneous drainage of pancreatic necrosis results in the evacuation of liquefied necrotic contents, leaving solid necrotic debris within the collection lumen [14]. Thus, the therapeutic process takes longer and requires necrosectomy procedures to be performed earlier and more frequently [14]. In addition, evacuation of liquefied necrotic contents via percutaneous drainage, with solid debris being retained within the collection, contributes to technical difficulties, and in some cases, it prevents the endoscopic formation of a transmural junction.

Since the first reports on the dual-modality drainage technique [56], the methods for endoscopic and percutaneous treatment of pancreatic necrosis have evolved toward a more aggressive [1,2,3,4,5,6,7,8,9,10,11,12,13,14,16,19,20,21,22,23,24,25,26,27,28,29,30,31,32,33,34,35,35,36,38,39,40,41,59], necrosectomy-based approach, which is also reflected in the aforementioned PEN approach, as well as in the multiple-access strategy in which PEN is combined with transmural endoscopic access techniques [37,42,43,44,45,46,47,48,49,50,51,52,53,54,55]. In more than one-third of patients who underwent PEN, percutaneous access was combined with transmural drainage techniques, which contributed to the efficacy of minimally invasive pancreatic necrosis management [37,42,43,44,45,46,47,48,49,50,51,52,53,54,55].

In cases where necrotic collections are far from the gastrointestinal tract wall, PEN may be used as an escalation technique following failure in securing retroperitoneal percutaneous access. In such cases, PEN also presents as an alternative to retroperitoneal surgical necrosectomy in the step-up approach. In cases of transmural endoscopic drainage for necrotic collections, characterized by the location of the main part of the collection outside the omental sac and the necrosis penetrating the paracolic gutters towards the pelvis, PEN preceded by percutaneous drainage increases the efficacy of endotherapy for the consequences of acute necrotizing pancreatitis [40,55].

Retroperitoneal abscesses are a subgroup of intraabdominal abscesses [60,61,62]. The recommended management strategy for intraabdominal abscesses consists of abdominal sepsis control [61,62], as can be achieved using minimally invasive treatment techniques. Efficient drainage is the first stage in the interventional management of intraabdominal abscesses [61,62]. In the event of inefficient drainage techniques, often due to the presence of large quantities of solid debris within the collection contents, the next treatment stage should be necrosectomy. In cases of retroperitoneal abscesses, this can be achieved via PEN. Thus, along with the minimally invasive treatment of pancreatic necrosis, PEN is also useful for the interventional treatment of retroperitoneal abscesses in daily clinical practice.

The PEN technique is significantly different from other previously reported techniques of surgical necrosectomy performed from retroperitoneal access using a flexible endoscope (sinus tract endoscopy) [63]. First, in our opinion, no mechanical broadening of the percutaneous access to the collection lumen is required in PEN, reducing the risk of persistent pancreaticocutaneous fistulas [37,42,43,44,45,46,47,48,49,50,51,52,53,54,55]. The edges of the drainage channel, as established in the course of sinus tract endoscopy, are frequently well defined due to multiple mechanical broadening procedures, which increases the risk of persistent pancreaticocutaneous fistula formation [21,22,23,24,25,27,28]. Second, the use of a fully coated esophageal stent ensures complete sealing of the channel between the necrotic collection and the skin coatings, reducing the risk of the collection contents leaking to other anatomical spaces and causing secondary infections. In addition, keeping the percutaneously introduced esophageal stent in place throughout the treatment provides wide access to the collection cavity, facilitating free flow and evacuation of its contents during the post-procedural period [37,42,43,44,45,46,47,48,49,50,51,52,53,54,55].

A common technical problem that is encountered during PEN is insufflation of the necrotic collection during a difficult procedure that is caused by a large esophageal stent diameter relative to the diameter of the endoscope. It appears that the sealing of the outer end of the esophageal stent using various silicone seals or laparoscopic reducers may be helpful in maintaining the proper pressure within the cavity lumen, thus, enabling sufficient visibility during the endoscopic surgery.

In our opinion, based on our experience, the most common complications of PEN include self-limited intra-cavity bleeding that does not require surgical intervention and pancreatic cutaneous fistulas that frequently resolve spontaneously without the requirement of additional medical interventions. The formation of pancreatic cutaneous fistulas is a very controversial aspect of percutaneous necrosectomy procedures, regardless of the technique. In PEN, all patients present with fistulas following the removal of the esophageal stents, although the fistulas usually resolve spontaneously within a few days. Only a persistent pancreatic cutaneous fistula can be considered a treatment complication.

The main limitations of our study are the small amount of data and small number of available articles about PEN in the current literature. Moreover, small studies and small study groups are more heterogeneous. If there is heterogeneity, treatment effects in individual studies may deviate more from the summary effect than expected by chance. The possible heterogeneity among studies on the definition of clinical success may also negatively impact the results of our review.

The study showed that the PEN technique is potentially effective, with an acceptable rate of complications, and may be implemented with good clinical results in patients with pancreatic necrosis. The problem with the exact determination of its clinical value, i.e., its efficacy and safety, is that the data were limited because they was derived from individual case reports [37,42,43,44,45,46,47,48,49,50,51,52,53,54,55]. Nonetheless, the outcomes of PEN are promising, as confirmed by our experience in our treatment facility.

## Data Availability

Data available on request (corresponding author email: matjagiel@gmail.com).

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
