# Peer review of "Percutaneous Endoscopic Necrosectomy—A Review of the Literature"

_jcm, 2022, doi:10.3390/jcm11143932_

Round 1

Reviewer 1 Report

The aim of the publication was to evaluate the role of percutaneous endoscopic necrosectomy (PEN) in the interventional treatment of pancreatic necrosis. 

The choice of intervention for necrotizing pancreatitis is an actual problem of modern surgery. 

A minimally invasive step-up approach is now preferred, involving endoscopic and or percutaneous catheter drainage as the first step, followed by endoscopic or minimally invasive surgical necrosectomy as required, is now preferred. 

The authors devoted a significant part of their review to this issue. Obviously, this shouldn't have been done. In addition, they used a limited number of current publications (over the last 5 years), which is a significant disadvantage.

The authors tried to present the role of PEN in the form of a systematic review. However, its structure does not comply with the principles of evidence-based medicine.

I strongly recommend that the authors adding and describing at least the following sections in the "Materials and Methods":

Literature search;

Eligibility criteria;

Data extraction and quality assessment;

Statistical analysis.

Author Response

Dear Reviewer,

First of all, I would like to thank you for positive review of our manuscript. According to Reviewer’s suggestion, we have made the appropriate corrections in the manuscript.

Reviewer #1

The aim of the publication was to evaluate the role of percutaneous endoscopic necrosectomy (PEN) in the interventional treatment of pancreatic necrosis. 

The choice of intervention for necrotizing pancreatitis is an actual problem of modern surgery. 

A minimally invasive step-up approach is now preferred, involving endoscopic and or percutaneous catheter drainage as the rst step, followed by endoscopic or minimally invasive surgical necrosectomy as required, is now preferred. 

The authors devoted a significant part of their review to this issue. Obviously, this shouldn't have been done. In addition, they used a limited number of current publications (over the last 5 years), which is a significant disadvantage.

Answer: Thank you for your comments. Yes, we agree with the Reviewer that we wrote a general and wide introduction into the issue of endoscopic and minimally invasive surgical/percutaneous  necrosectomy. In our opinion it is a natural method to introduce the Reader to the issue of percutaneous endoscopic necrosectomy (PEN), which combines methods of endoscopic and surgical (percutaneous) necrosectomy. In our paper we attempted to write a general and wide review of both older and current publications on this issue, especially the most important ones. We hope that this explanation will be accepted by Reviewers and Editors. As the Reviewer suggested, we added a few important publications from recent years.  We hope all the corrections will be satisfactory for the Reviewers and Editors and will make our paper fully understandable.

Reviewer #1

The authors tried to present the role of PEN in the form of a systematic review. However, its structure does not comply with the principles of evidence-based medicine. I strongly recommend that the authors adding and describing at least the following sections in the "Materials and Methods":

Literature search;

Eligibility criteria;

Data extraction and quality assessment;

Statistical analysis.

Answer: Thank you for your comments As the Reviewer suggested, we made changes in the "Materials and Methods” section in our manuscript. We added the mentioned subsections. We hope our corrections will be useful for the readers.

We hope that made corrections will satisfy both the Reviewers and the Editors.

We hope that our corrections will make the manuscript meet the requirements for publication in “Journal of Clinical Medicine”.

With kind regards,

Mateusz Jagielski, Agata Chwarścianek, Jacek Piątkowski and Marek Jackowski

Reviewer 2 Report

In this work, Jagielski and colleagues reported a review of the available literature on percutaneous endoscopic necrosectomy (PEN) using esophageal self-expanding metal stents (SEMS). The Authors reviewed various techniques of pancreatic WON drainage and necrosectomy, and subsequently focused on PEN, reporting outcomes of 52 procedures from 15 studies. Finally, they concluded that PEN is highly effective minimal invasive method of 15 treatment of pancreatic necrosis, with an acceptable rate of complication.

This work has the merit of having reported an extensive review of the literature about endoscopic, percutaneous and surgical necrosectomy. However, the study design and the structure of the manuscript are ambiguous, and the work needs profound changes before being considered for publication.

Major points:

-        It is unclear if the primary aim of the study is to review the literature on necrosectomy or specifically discuss PEN with esophageal SEMS. In the first case, the manuscript should be structured as a narrative review. In the second case, the introduction should be largely reduced, and PEN should be discussed more in depth, reporting more details about the cited studies, indication of the procedures, possible concomitant/adjunctive drainage procedures, and detailed review of adverse events.

-        According to the study design, the title and the abstract should be modified accordingly.

Minor points:

-        Some errors are present in the reference list (eg, ref 40 and 55 are the same work).

Author Response

Dear Reviewer,

First of all, I would like to thank you for positive review of our manuscript. According to Reviewer’s suggestion, we have made the appropriate corrections in the manuscript.

Reviewer #2

In this work, Jagielski and colleagues reported a review of the available literature on percutaneous endoscopic necrosectomy (PEN) using esophageal self-expanding metal stents (SEMS). The Authors reviewed various techniques of pancreatic WON drainage and necrosectomy, and subsequently focused on PEN, reporting outcomes of 52 procedures from 15 studies. Finally, they concluded that PEN is highly effective minimal invasive method of 15 treatment of pancreatic necrosis, with an acceptable rate of complication.

This work has the merit of having reported an extensive review of the literature about endoscopic, percutaneous and surgical necrosectomy. However, the study design and the structure of the manuscript are ambiguous, and the work needs profound changes before being considered for publication.

Answer: Thank you very much for your positive review. We will make every effort to make necessary changes in the manuscript to increase its quality.

Reviewer #2

Major points:

-        It is unclear if the primary aim of the study is to review the literature on necrosectomy or specifically discuss PEN with esophageal SEMS. In the first case, the manuscript should be structured as a narrative review. In the second case, the introduction should be largely reduced, and PEN should be discussed more in depth, reporting more details about the cited studies, indication of the procedures, possible concomitant/adjunctive drainage procedures, and detailed review of adverse events.

Answer: Thank you for your comment. The primary aim of our study is to review the literature on necrosectomy with emphasis on percutaneous endoscopic necrosectomy (PEN) using esophageal self-expanding metal stents (SEMS). We added this aim to the text of our paper. As the Reviewer suggested, we re-structed the manuscript in form of comprehensive review of current literature about this issue. We re-structured mainly the “Materials and Methods” section. We hope all the corrections will be satisfactory for the Reviewers and Editors and will make our paper fully understandable.

-        According to the study design, the title and the abstract should be modified accordingly.

Answer: As the Reviewer suggested, we changed the title and re-structured the abstract of our manuscript. We hope our corrections will be useful for the readers.

Minor points:

-        Some errors are present in the reference list (eg, ref 40 and 55 are the same work).

Answer: Thank you for your comment. We corrected that error. Moreover, we corrected the “Reference” section.

We hope that made corrections will satisfy both the Reviewers and the Editors.

We hope that our corrections will make the manuscript meet the requirements for publication in “Journal of Clinical Medicine”.

With kind regards,

Mateusz Jagielski, Agata Chwarścianek, Jacek Piątkowski and Marek Jackowski

Reviewer 3 Report

This is a review of the treatment of pancreatic necrosis with a focus on percutaneous endoscopic necrosectomy through a self-expandable esophageal stent (PEN). I find this to be a well written paper, however, I do have some questions and comments.

All of the studies included in the review of the PEN procedure where case reports or small retrospective cohorts except for the 9-patient prospective cohort in the study by Saumoy et al. It is possible that there could be tendency to report cases that has gone well and an underreporting of cases with severe complications. Although the outcomes for the patients included in this review of the PEN procedure is encouraging I do not think it is accurate to state that the PEN technique is highly effective, with an acceptable rate of complications since the true rate of favorable outcomes and complications is not known.

It is stated that no mechanical broadening of the percutaneous access to the collection lumen is required in PEN, reducing the risk of persistent pancreaticocutaneous fistulas with reference to the studies included in the review of the PEN procedure. These studies show that no mechanical broadening of the percutaneous access to the collection lumen is required in PEN. However, I cannot see that they directly prove that this leads to a reduced risk of persistent pancreaticocutaneous fistulas. The next sentence states that the multiple mechanical broadening procedures of the drainage channel increases the risk of persistent pancreaticocutaneous fistula formation. This seems like a likely hypothesis but I do not see that it is really proven be the studies referred to. Please clarify.

It is stated that the most common complications of PEN include self-limited intra-cavity bleeding that does not require surgical intervention and pancreaticocutaneous fistulas that frequently resolve spontaneously without the requirement of additional medical interventions. However, one of the references for this statement seems to be about sinus tract endoscopy (referens 60) and the other is missing from the reference list (referens 61).

Author Response

Dear Reviewer,

First of all, I would like to thank you for positive review of our manuscript. According to Reviewer’s suggestion, we have made the appropriate corrections in the manuscript. In the further paragraphs of this response we would like to address each every sentence of the review and thoroughly respond to Reviewer’s suggestion.

Reviewer #3

This is a review of the treatment of pancreatic necrosis with a focus on percutaneous endoscopic necrosectomy through a self-expandable esophageal stent (PEN). I find this to be a well written paper, however, I do have some questions and comments.

Answer: Thank you very much for your positive review.

 Reviewer #3

All of the studies included in the review of the PEN procedure where case reports or small retrospective cohorts except for the 9-patient prospective cohort in the study by Saumoy et al. It is possible that there could be tendency to report cases that has gone well and an underreporting of cases with severe complications. Although the outcomes for the patients included in this review of the PEN procedure is encouraging I do not think it is accurate to state that the PEN technique is highly effective, with an acceptable rate of complications since the true rate of favorable outcomes and complications is not known.

Answer: We fully agree with the Reviewer. Our aim was to show that PEN technique is possible to implement with good clinical effect. We fully agree with the Reviewer, that basing only on current literature it is difficult to draw final conclusions. In the paper, we also presented our experience with this technique in form of “Algorithm of therapeutic management of patients with pancreatic necrosis” and figures from procedures. In compliance with the above, we corrected conclusions in our article. We hope that this explanation will be accepted by Reviewers and Editors. We hope all the corrections will be satisfactory for the Reviewers and Editors and will make our paper fully understandable.

Reviewer #3

It is stated that no mechanical broadening of the percutaneous access to the collection lumen is required in PEN, reducing the risk of persistent pancreaticocutaneous fistulas with reference to the studies included in the review of the PEN procedure. These studies show that no mechanical broadening of the percutaneous access to the collection lumen is required in PEN. However, I cannot see that they directly prove that this leads to a reduced risk of persistent pancreaticocutaneous fistulas. The next sentence states that the multiple mechanical broadening procedures of the drainage channel increases the risk of persistent pancreaticocutaneous fistula formation. This seems like a likely hypothesis but I do not see that it is really proven be the studies referred to. Please clarify.

Answer: Thank you for your comments. In our opinion, no mechanical broadening of the percutaneous access to the collection lumen with use of high-pressure balloon is required in PEN, reducing the risk of persistent pancreaticocutaneous fistulas. The edges of the drainage channel as established in the course of sinus tract endoscopy are frequently well defined due to multiple mechanical broadening procedures, which increases the risk of persistent pancreaticocutaneous fistula formation. The formation of pancreaticocutaneous fistulas is a very controversial aspect of percutaneous necrosectomy procedures, regardless of the technique. In PEN, all patients present with fistulas following the removal of the esophageal stents, although the fistulas usually resolve spontaneously within a few days. Only a persistent pancreaticocutaneous fistula can be considered a treatment complication. In the text of our paper, we emphasize that this is only our opinion, difficult to prove basing on available literature.  We hope that this explanation will be accepted by Reviewers and Editors. We hope all the corrections will be satisfactory for the Reviewers and Editors and will make our paper fully understandable.

Reviewer #3

It is stated that the most common complications of PEN include self-limited intra-cavity bleeding that does not require surgical intervention and pancreaticocutaneous fistulas that frequently resolve spontaneously without the requirement of additional medical interventions. However, one of the references for this statement seems to be about sinus tract endoscopy (referens 60) and the other is missing from the reference list (referens 61).

Answer: Thank you for your comments. We corrected that error. Moreover, we corrected the “Reference” section. We corrected the text of our paper to make it more clear and readable.

In conclusion, we hope that made corrections will satisfy both the Reviewers and the Editors.

We hope that our corrections will make the manuscript meet the requirements for publication in “Journal of Clinical Medicine”.

With kind regards,

Mateusz Jagielski, Agata Chwarścianek, Jacek Piątkowski and Marek Jackowski

Round 2

Reviewer 1 Report

I have no comments.

Author Response

Dear Reviewer,

We would like to thank you for positive review of our manuscript.

Reviewer #1

I have no comments.

Answer: Thank you for positive review of our manuscript.

With kind regards,

Mateusz Jagielski, Agata Chwarścianek, Jacek Piątkowski and Marek Jackowski

Reviewer 2 Report

In the revised version of the manuscript, changes have been made according to comments

I suggest these further changes:

I would make clear in the abstract and in the Results paragraph that the described results come from 15 studies after review of the literature

I suggest summarizing the results in a table, for example expanding table 1. Importantly, data on adverse events (and severe adverse events), technical success, clinical success and concomitant therapies (endoscopic transmural or surgical drainage) should be reported. Moreover, possible heterogeneity among studies on definition of clinical success should be discussed.

Author Response

Dear Reviewer,

First of all, we would like to thank you for positive review of our manuscript. According to Reviewer’s suggestion, we have made the appropriate corrections in the manuscript.

Reviewer #2

In the revised version of the manuscript, changes have been made according to comments

Answer: Thank you very much for your positive review.

Reviewer #2

I suggest these further changes:

Answer: Thank you for your comments. We will make every effort to make necessary changes in the manuscript to increase its quality.

Reviewer #2

I would make clear in the abstract and in the Results paragraph that the described results come from 15 studies after review of the literature.

Answer: As the Reviewer suggested, we re-structured the abstract of our manuscript. We added this information to the abstract of our paper. We hope our corrections will be useful for the readers.

Reviewer #2

I suggest summarizing the results in a table, for example expanding table 1. Importantly, data on adverse events (and severe adverse events), technical success, clinical success and concomitant therapies (endoscopic transmural or surgical drainage) should be reported.

Answer: Thank you for your comment. We re-structured Table 1. We added mentioned-above results from studies to Table 1 of our manuscript. We hope all the corrections will be satisfactory for the Reviewers and Editors and will make our paper fully understandable.

Reviewer #2

Moreover, possible heterogeneity among studies on definition of clinical success should be discussed.

Answer: Thank you for your comment. We corrected the Discussion section. We added information about limitations of our paper. Moreover, we discussed about possible heterogeneity among studies on definition of clinical success. We hope our corrections will be useful for the readers.

We hope that made corrections will satisfy both the Reviewers and the Editors.

We hope that our corrections will make the manuscript meet the requirements for publication in “Journal of Clinical Medicine”.

With kind regards,

Mateusz Jagielski, Agata Chwarścianek, Jacek Piątkowski and Marek Jackowski